# The Cross-Spillover Effects of Online Prosocial Behavior on Subjective Well-Being: Daily Diary Evidence from Chinese Adolescents

**Weida Zhang [1,†], Guoliang Yu [1,†] and Wangqian Fu [2,\*]**

[1]  School of Education, Renmin University of China, 59 Zhongguancun Ave., Haidian District, Beijing 100872, China
[2]  Faculty of Education, Beijing Normal University, 19 Xinjiekouwai Ave., Haidian District, Beijing 100875, China
\*  Correspondence: fu2021@bnu.edu.cn
†  These authors contributed equally to this work.

**Abstract:** This study investigated the effects of online prosocial behavior on the subjective well-being of adolescents and its spillover and crossover effects. By convenience sampling, this paper adopted a diary survey method to collect the daily online prosocial behavior and subjective well-being data of 120 first-grade junior high school students and their parents for 5 consecutive days during their winter vacation in China. The online prosocial behaviors of adolescents during the day can significantly positively predict their subjective well-being during the day and at night, which indicates that adolescents' subjective well-being has a spillover effect from online to offline. In addition, online prosocial behavior and the subjective well-being of adolescents in the daytime are significantly positively correlated with their parents' subjective well-being at night, indicating that there is a crossover effect between online prosocial behavior and the subjective well-being of adolescents in the daytime and their parents' subjective well-being at night. It is important to create a good online environment for adolescents and promote the benign spillover and crossover effect of online prosocial behavior on subjective well-being.

**Keywords:** adolescents; online prosocial behavior; subjective well-being; spillover effect; crossover effect

## 1. Introduction

According to the Report on China's Internet Development (2021) released by the Internet Society of China, there were 989 million Internet users by the end of 2020 in China, and the Internet penetration rate had reached 70.4%. In particular, the total number of mobile Internet users exceeded 1.6 billion people. Young people are enthusiastic users of the Internet. According to the statistics on Internet use in 2020 by Eurostats, about 96% of young people use the Internet every day [1]. The widespread use of the Internet has brought great convenience to human social activities; at the same time, it has also caused many problems, such as Internet addiction, bullying, fraud, and so on. A cross-country study involving nine European countries found that time spent online was associated with problematic Internet use, with the greatest prevalence in teenagers' among problematic Internet users, ranging from 14.3% to 54.9% [2].

Teenagers are in a critical period of physical and mental development, and they are vulnerable to mental health problems, which leads to concerns from families, schools, and people from all walks of life about teenagers' use of the Internet. Meanwhile, adolescents' prosocial behaviors in the use of the Internet are universal, and the incidence of prosocial behaviors is higher than the frequency of antisocial behaviors [3]. Studies have found that there is a significant positive correlation between online prosocial behaviors and offline prosocial behaviors, and both are significantly negatively correlated with online problem

behavior [4]. Apart from these behaviors, there is a close relationship between social media use and subjective well-being indicators, such as life satisfaction [5], depression [6], and emotional well-being [7].

Online prosocial behavior refers to the behaviors that individuals voluntarily carry out in the electronic environment to benefit specific people or promote harmonious relations with others [8]. There are many types of online prosocial behaviors, including comforting friends online, sharing resources and information with classmates online, and helping peers on social networking sites. At present, only a few studies have explored the influencing factors of online prosocial behavior [8], and the studies targeting adolescents are even rarer. Existing studies have found that females report more online prosocial behaviors than males [9] and that there is a significant positive correlation between online prosocial behaviors and offline prosocial behaviors [4]. Individuals' frequency of use of electronic media and their emotional state [8] are correlated with online prosocial behaviors.

Subjective well-being refers to an individual's evaluation of his/her life's cognitive and emotional status [10]. As a multidimensional concept, it is composed of life satisfaction, positive emotion, and negative emotion. Adolescent subjective well-being is an important factor influencing academic and interpersonal development; it is an important index by which to measure the success of school education. Prosocial behavior has obvious benefits for the positive development and mental health of adolescents [11]. Existing studies have shown that there is a significant positive correlation between individuals' offline prosocial behaviors and subjective well-being [12,13]. According to the basic psychological needs satisfaction theory, individuals have three innate basic psychological needs, namely, relational needs, ability needs, and autonomy needs. When basic psychological needs are satisfied, individual subjective well-being will be improved [13]. In a 14-day diary study, it was found that individual daily prosocial activities (for example, taking the initiative to care about peers' emotions and opening the door for others) were significantly associated with an increased level of positive emotion and mental health, and the relationship between the daily prosocial activities and the level of negative emotion was not significant [14]. According to the additive interaction model [15], well-being can be generated through the interaction between people and their environment. Online prosocial behavior is a form of interaction between individuals and the online environment. Grossman et al. analyzed data from collated MIDUS II study diary surveys and found that when individuals participated in formal volunteering or provided informal help, they experienced greater self-promotion (pride, confidence) and social connection (closeness to others, the feeling of belonging to the group), etc. [16]. "Pursue self and be true to yourself" is the watchword of many teenagers [17]. In offline social activities, teenagers may not be able to express themselves truthfully under pressure, while the Internet environment facilitates the development of authenticity for teenagers. Psychological theory suggests that authenticity is the decisive factor in subjective well-being. Existing studies have found that individual authenticity and its associated subjective experience of self-integrity and consistency can increase self-satisfaction and satisfy basic psychological needs, thereby promoting positive emotions and reducing negative emotions [17].

There are dozens of studies that have explored the relationship of online prosocial behavior with the subjective well-being of adolescents. Most of them were conducted using a questionnaire survey, which contributes to understanding the level of subjective well-being, although it is limited in terms of the effect of that influence across time and individuals. Previous studies have mostly used inter-subject study designs (using more than one group of subjects to estimate the effects of experimental treatments, by comparing two or more groups of subjects) and, thus, have not been able to explain whether these variables work on individuals rather than among a group. For example, if researchers are trying to determine whether there is a relationship between individual prosocial behavior and subjective well-being later in the day, then intra-subject data analysis is needed. Diary research allows researchers to take multiple measurements of the same subject at multiple points in a day, so it is suitable for data analysis between subjects [18]. In view of this

ability, this study used a diary survey method to measure the online prosocial behavior of adolescents in a particular week. Conducted in a natural setting, diary surveys allow repeated measurements of subjects over time to capture events and experiences. This type of data collection facilitates study designs in which one group of subjects receives all experimental processing. In addition, it greatly shortens the time span between the event or experience and the individual report, thus reducing the recall bias problem. Although the online activity of adolescents has drawn attention in China, few studies have investigated the effects of adolescents' online prosocial behavior on their subjective well-being.

To address these gaps in the literature, based on the additive interaction model and the previous research evidence [16,17], the current study applied a diary survey method to study the online prosocial behavior and subjective well-being of selected adolescents. Meanwhile, we also investigated their offline prosocial behavior and the subjective well-being of their parents in China. These factors can contribute to reducing the recall bias problem of studying prosocial behavior and the subjective well-being of adolescents. Moreover, we included both the adolescents and their parents in the study to analyze the crossover effects, which are significant in terms of family dynamics.

There were three hypotheses in the study:

**Hypothesis 1 (H1).** *Online prosocial behavior predicts adolescents' subjective well-being.*

**Hypothesis 2 (H2).** *The subjective well-being of adolescents during the daytime can predict their subjective well-being at bedtime of Chinese adolescents.*

**Hypothesis 3 (H3).** *Chinese adolescents' daytime subjective well-being can predict their parents' nighttime subjective well-being.*

## 2. Method

The present study used an intensive longitudinal design. Enrolled participants completed a baseline survey, followed by 5 days of daily diary surveys about their online prosocial behavior and subjective well-being.

### 2.1. Participants

The subjects of this study were grade-one students of a middle school in Nanjing and their parents. We contacted the principals of the middle school to obtain their permission and support. Then, the head teachers of the grade-one middle school helped to recruit the voluntary students and obtained permission from the students' parents to participate in the study. Of 171 students who participated in the initial survey, 133 agreed to participate in a follow-up daily diary survey, measuring participants' daily online prosocial behavior and daily subjective well-being. The incomplete data were not included in the analysis, since some of the adolescents did not complete the diary survey at the specified time, as required by the study, or did not interact well with their parents on that day. Finally, valid data from 120 middle-school students and their parents had been obtained.

### 2.2. Procedure

Diary investigation under natural circumstances is a research method that can reproduce daily life [18]. The survey data of this study were collected by means of a questionnaire during the summer vacation. Respondents first answered demographic questions. Then, the researchers collected online prosocial behavior and subjective well-being data from each subject over five consecutive working days. With reference to the relevant research and students' school schedules in Nanjing middle schools, this study required adolescents to complete an online prosocial behavior and subjective well-being diary investigation at 5 p.m. every day (T1); then, at 10 p.m., adolescents and their parents each completed a once-daily self-report of subjective well-being (T2). The final data set consisted of 2400 diary responses from 360 participants (120 middle-school students and their parents). Finally,

the researchers analyzed the daily survey data through a multi-level structural equation model (MSEM).

### 2.3. Measurements

The online prosocial behavior scale (OPBS), compiled and simplified by Erreygers et al. (2018), was used to conduct a diary survey of adolescents' online prosocial behavior. According to the suggestion made by Erreygers et al. (2018) [3], 5 of the questions (e.g., "I praised someone else online today"; "I helped someone else online today") were selected to reduce the burden on the subjects to find the appropriate words for an answer, thus ensuring the reliability of the data. Online prosocial behavior was scored on a Likert scale of 1 (none at all) to 5 (very many), with higher scores indicating more instances of online prosocial behavior. Cronbach's alpha coefficient was calculated separately for each day and ranged from 0.82 to 0.95, giving a mean of 0.91.

Subjective well-being can be measured by a single question [7]. In the measurement of subjective well-being in this study, each subject was required to answer the question "On the whole, I feel very happy at present". The scale was rated on the Likert scale from 1 (not at all) to 7 (very strong); the higher the score, the higher the subjective well-being of the teenagers. To confirm the convergent validity of this item, a previous pilot daily survey among 32 adolescents was conducted by the first author. The subjective well-being item was strongly correlated with positive affect ($r = 0.76$, $p < 0.001$) and negative affect ($r = -0.71$, $p < 0.001$).

### 2.4. Data Analysis

Since the repeated measurements obtained by the diary survey were nested within individuals, considering the multilevel structural equation model (MSEM) can analyze the within and between paths of all variables [19]. Mplus 8.0 [20] was used to conduct an MSEM to test the mediating effects [19]. To test T1 adolescents' online prosocial behavior, T1 adolescents' subjective well-being was used to predict T2 adolescents' and parents' subjective well-being. Among them, the predictive variables (T1—adolescent happiness, T1—maternal happiness, and T1—paternal happiness), the mediator variable (T2 adolescent happiness), and the outcome variable (T2—adolescent online prosocial behavior) are all above the same level. Before operating the MSEM analysis, we calculated the correlation between variables; all correlations were below $r < 0.50$, indicating that there were no multicollinearity problems in the present study. Meanwhile, we tested for potential common method bias (CMB) from a statistical perspective based on Harman's single-factor test. The first unrotated factor captured only 29% of the variance in data, suggesting that the results were not biased based on CMB.

## 3. Results

### 3.1. Online Prosocial Behavior of Adolescents Has a Positive Influence on Subjective Well-Being

Pearson correlation analysis was applied to test the correlation between participants' online prosocial behavior and subjective well-being. The results of the mean value, standard deviation, and correlation coefficient of multiple observational variables of the subjects were shown in Table 1. The T1 subjective well-being of adolescents' daytime subjective well-being ($r = 0.42$, $p < 0.001$), the T2 subjective well-being of adolescents' nighttime subjective well-being ($r = 0.21$, $p < 0.001$), the T2 subjective well-being of paternal nighttime subjective well-being ($r = 0.15$, $p < 0.001$), and the T2 subjective well-being of maternal nighttime subjective well-being ($r = 0.18$, $p < 0.001$) were significantly positively correlated with T1 online prosocial behavior in terms of adolescents' daytime prosocial behavior.

**Table 1.** The mean value, standard deviation, and correlation coefficients of variables.

| Variables | M | SD | 1 | 2 | 3 | 4 | 5 |
|---|---|---|---|---|---|---|---|
| Adolescents' T1 OPB | 9.92 | 4.64 | 1 | | | | |
| Adolescents' T1 SWB | 3.56 | 1.08 | 0.46 * | 1 | | | |
| Adolescents' T2 SWB | 3.46 | 1.08 | 0.23 *** | 0.71 *** | 1 | | |
| Fathers' T2 SWB | 3.02 | 1.03 | 0.09 | 0.34 *** | 0.40 *** | 1 | |
| Mothers' T2 SWB | 3.12 | 1.01 | 0.11 | 0.39 *** | 0.40 *** | 0.62 *** | 1 |

Note. OPB is online prosocial behavior, SWB is subjective well-being, * $p < 0.05$; *** $p < 0.001$.

### 3.2. Spillover Effects of Online Prosocial Behavior on Subjective Well-Being

In terms of the spillover hypothesis (H2), an MSEM was applied to test the direct relationship between T1 and T2 regarding the subjective well-being of adolescents (see Table 2). Whether within or among individuals, T1 subjective well-being had a significant positive impact on T2 subjective well-being ($p < 0.001$); therefore, research Hypothesis H2 was verified. In other words, adolescents who reported higher levels of subjective well-being during the day also reported higher levels of subjective well-being that evening (intra-individual). Individuals who self-reported that their subjective well-being level was higher than average during the day also had a subjective well-being level that was higher than average at night (inter-individual).

**Table 2.** Direct effects of multiple 1-1-1 MSEM mediation models with a fixed slope.

| | Intrapersonal Level | | | Interindividual Level | | |
|---|---|---|---|---|---|---|
| | b | SE | p | b | SE | p |
| T1 SWB | | | | | | |
| T1 OPB | 0.286 | 0.071 | 0.000 | 0.295 | 0.065 | 0.000 |
| T2 SWB | | | | | | |
| T1 OPB | 0.228 | 0.198 | 0.000 | 0.128 | 0.084 | 0.000 |
| T1 SWB | 0.119 | 0.291 | 0.000 | 0.021 | 0.216 | 0.000 |
| T2 Fathers' SWB | | | | | | |
| T1 OPB | | | | 1.046 | 2.311 | 0.000 |
| T1 SWB | | | | 1.455 | 0.734 | 0.000 |
| T2 SWB | | | | 2.331 | 0.066 | 0.000 |
| T2 Mothers' SWB | | | | | | |
| T1 OPB | | | | 1.362 | 2.156 | 0.000 |
| T1 SWB | | | | 1.615 | 0.334 | 0.000 |
| T2 SWB | | | | 2.552 | 0.145 | 0.000 |

### 3.3. Cross-Effects of Online Prosocial Behavior on Subjective Well-Being

To test the cross-hypothesis (H3), we examined the association between adolescent T1 subjective well-being and their parents' T2 subjective well-being (see Table 2). The results showed that adolescents' daytime subjective well-being could positively predict their parents' nighttime subjective well-being ($p < 0.001$). Therefore, hypothesis H3 was found to be valid.

### 4. Discussion

In this study, we conducted a five-day diary survey of 120 families in Nanjing during the summer vacation, to explore the influence of daily online prosocial behavior on adolescents' daytime subjective well-being and whether it can predict the adolescents' and their parents' nighttime subjective well-being.

First, we found that the online prosocial behavior of adolescents was significantly positively correlated with subjective well-being during the day and at night, and that online prosocial behavior could significantly predict subjective well-being, which was consistent with the findings of Erreygers et al. (2017) [8]. Previous studies on offline prosocial behaviors showed that individual prosocial behaviors are significantly positively correlated

with their subjective well-being [21]. That is, prosocial behavior brings many benefits not only to the recipient but also to the actor. Prosocial behavior can promote the level of subjective well-being by promoting reciprocity and social integration, as well as the sense of competency and meaning of prosocial behavior practitioners in life [22]. The results of this study indicate that the same conclusion obtained from offline prosocial behavior is also applicable to the relationship between online prosocial behavior and subjective well-being. The possible explanation for the connection between prosocial behavior and subjective well-being is that prosocial behavior can bring about the satisfaction of basic psychological needs (autonomy, relationality, and competency needs) from the point of view of the actor, and the satisfaction of basic psychological needs can improve the subjective well-being of the individual.

Our results also found that adolescents who self-reported higher than average subjective well-being during the day also reported higher than average subjective well-being at night, which indicates that adolescents' subjective well-being has a spillover effect from online to offline. It shows that adolescents' subjective well-being is stable, which is consistent with the results of previous relevant research literature [23].

In addition, online prosocial behavior and the subjective well-being of adolescents in the daytime are significantly positively correlated with their parents' subjective well-being at night, which indicates that there is a crossover effect between online prosocial behavior and the subjective well-being of adolescents in the daytime and their parents' subjective well-being at night. This is consistent with the results of a study by Matjasko and Feldman in 2005 [24]. They found that the happiness of mothers returning home from work was transmitted to their adolescent children, while there was no significant crossover effect between the emotions of fathers and adolescent children [24]. This phenomenon may be explained by the fact that mothers are usually more involved in their children's emotional life than fathers are [25]. Fathers are generally unable to effectively transfer positive emotions to their children. This may be due to the fathers spending less time with their children on average [26]. This finding extends research on the crossover effects of emotions. On the one hand, the existing studies on the crossover effects of emotions mainly focus on negative emotions (such as stress, anxiety, etc.), while there are relatively few studies on whether there is a cross-cutting effect with positive emotions. On the other hand, existing research objects mainly focus on the husband–wife relationship among family members, and there are still few studies on whether the crossover effects of emotions can be applied between parents and children. In particular, relevant studies have been carried out, mainly in the context of Western culture, and the results of this study answer the question of whether these findings can be verified in the context of Chinese culture.

## 5. Limitations and Further Research

Although the hypothesis of this study has been verified, there are still some limitations. First, in this study, only the spillover and crossover effects of the subjective well-being of the young people were investigated; the effects in terms of gratitude, optimism, and self-compassion were not assessed. Prosocial behavior and subjective well-being have the problem of social desirability; participants are likely to answer questions with deliberately positive answers. In future studies, controls for social desirability could be added, and the spillover and crossover effects of other positive emotions can also be explored.

Secondly, although we collected data by referring to the number of subjects in previous diary survey methods, the sample size selected in this study was relatively small, so the statistical power was limited. In future studies, a larger sample can be selected to examine the impact of online prosocial behaviors more effectively on subjective well-being and whether spillover and crossover effects will occur.

Thirdly, existing research results show that factors such as parent-child relationship quality and density are affected to different extents due to the level of daily effective interaction between parents [27]; this was considered in this study in the regulation of these

variables. In future studies, these factors can be included in the control variable or variables for analysis.

Finally, although a diary method is statistically valid, the present findings are based on correlational data. Thus, further experiments are needed to provide causal evidence in the future.

### 6. Conclusions and Implications

This study is an attempt to examine the spillover and crossover effects of positive outcomes (the subjective well-being of adolescents and their parents) of positive events (the online prosocial behavior of adolescents). The results show that online prosocial behavior has a positive impact on subjective well-being, and there is a cross- and spillover effect. In other words, online prosocial behavior can bring instant subjective well-being to the participants and can positively predict the level of subjective well-being of adolescents and their parents at night.

The wide use of online multimedia facilitates the online prosocial behaviors of adolescents. The results of this study show that the online prosocial behaviors of adolescents can not only bring about benefits to the recipients but can also improve the subjective well-being level of the perpetrators. In addition, such positive emotional effects also have spillover and crossover effects. Therefore, it will be an important future research topic to encourage teenagers to use Internet resources correctly. Creating a good online environment for them and promoting the benign spillover and crossover effect of online prosocial behavior and subjective well-being are of great importance.

**Author Contributions:** Conceptualization, W.Z. and G.Y.; methodology, W.Z. and W.F.; data collection and analysis, W.Z. and W.F.; writing—original draft preparation, W.Z. and W.F.; writing—review and editing, W.F.; funding acquisition, W.F. All authors have read and agreed to the published version of the manuscript.

**Funding:** This research is supported by Youth project of Humanities and Social Sciences of the Ministry of Education (20YJC880015).

**Institutional Review Board Statement:** The studies involving human participants were reviewed and approved by the Ethics Committee of Renmin University of China.

**Informed Consent Statement:** Informed consent was obtained from all subjects involved in the study.

**Data Availability Statement:** The datasets generated and analyzed during the current study are available from the corresponding author on reasonable request.

**Conflicts of Interest:** The authors declare that they have no conflict of interest.

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
