# Peer review of "The Cross-Spillover Effects of Online Prosocial Behavior on Subjective Well-Being: Daily Diary Evidence from Chinese Adolescents"

_sustainability, doi:10.3390/su14159734_

Round 1

Reviewer 1 Report

This is a very good research. This will highlight better if you consider the following aspects of my suggestion.

1. Make the title more catchy one by highlighting the spillover effect.

2. Abstract needs to bring in some key aspects of  research focus as well in the beginning.

3. After the introduction there is a need to bring in research framework with a conceptual framework. That should bring the hypothesis in better positioned.

4. Better provide an overall focus of the methods and relevant methodology with citations at the beginning of methods section.

5. It is better to provide questions used in questionnaire or a brief description of it. This is missing.

6. Before discussions it is better to provide a  conclusion with some key dot points, that will lead discussions with backing of literature.

Author Response

Thank you very much for your comments and professional advice. These opinions help to improve academic rigor of our article. Based on your suggestion and request, we have made corrected modifications on the revised manuscript. Meanwhile, the manuscript had been reviewed and edited by language services. Furthermore, we would like to show the details as follow: 

Reviewer 2 Report

Dear author(s),

Regarding the submission "The influence of online prosocial behavior on the subjective well-being of Chinese adolescents and its cross-spillover effects", please find my comments below.

The article addresses a very important topic and presents a sufficient background, including references relevant to the research. Moreover, research design sounds appropriate, but the description of methods can be improved.

Regarding the methodology, the use of the Likert scale leads to collected data that do not meet the assumption of a continuous nature. Therefore, non-parametric techniques would be a better alternative to statistical analysis. Furthermore, considering the small sample size, it is also necessary to verify whether any deviations from normality may affect the results of the model applied in the study (MSEM).

The adopted correlation coefficient must also be described and justified (Pearson's R, Spearman's Rho, Kendall's Tau, other ?) For instance, if there is no symmetry in the distribution of the data, Pearson is not appropriate. In other cases, when one or more of the variables has outliers, the Kendall's Tau measure must be considered.

From the foregoing, I recommend accepting the paper after review (corrections/clarifications on these methodological aspects).

Sincerely.

Author Response

(The authors gave the same response as above.)

Reviewer 3 Report

sustainability- 1836545

The influence of online prosocial behavior on subjective well-being of Chinese adolescents and its cross-spillover effects

Weida et al.

 Overview and general recommendation:

The article is well written and the presentation is nice. I recommend a minor revision of the manuscript since the methods have not been sufficiently detailed. Provide a better explanation of the methods used to derive the results of Tables 1 and 2.

Author Response

(The authors gave the same response as above.)
